# Health-Related Quality of Life of Bone and Soft-Tissue Tumor Patients around the Time of Diagnosis

**DOI:** 10.3390/cancers15102804

**Published:** 2023-05-17

**Authors:** Anouk A. Kruiswijk, Desiree M. J. Dorleijn, Perla J. Marang-van de Mheen, Michiel A. J. van de Sande, Leti van Bodegom-Vos

**Affiliations:** 1Department of Biomedical Data Sciences, Medical Decision Making, Leiden University Medical Center, Albinusdreef 2, 2333 ZA Leiden, The Netherlands; 2Department of Orthopedic Surgery, Leiden University Medical Center, Albinusdreef 2, 2333 ZA Leiden, The Netherlands; 3Orthopedic Surgery, Ghent University Hospital, Corneel Heymanslaan 10, 9000 Gent, Belgium

**Keywords:** bone tumor, soft-tissue tumor, sarcoma, health-related quality of life, patient-reported outcomes, clinically relevant symptoms and limitations, diagnosis

## Abstract

**Simple Summary:**

Bone and soft-tissue tumor patients experience long-lasting physical and psychological challenges. However, it is unknown what health-related quality of life (HRQoL) challenges patients experience around the time of diagnosis and which group of patients are particularly affected. This study assessed the HRQoL of 637 patients around the time of diagnosis and compared their HRQoL scores to the general population. Overall, patients experienced increased levels of anxiety and pain and limitations in physical and social functioning compared to the general population. The increased level of pain and the limitation in physical functioning were clinically relevant. HRQoL differed between diagnosis subgroups: i.e., patients with malignant tumors experienced higher anxiety levels and pain, whereas patients with bone tumors had worse physical functioning compared with the general population. This indicates that the HRQoL is already affected around the time of diagnosis and should be taken into account to improve the quality of care delivered.

**Abstract:**

Background: Bone and soft-tissue tumor patients experience long-lasting physical and psychological challenges. It is unknown to what extent Health-Related Quality of Life (HRQoL) is already affected during the diagnostic process. This study assesses the HRQoL of bone and soft-tissue tumor patients around time of diagnosis and explores which patient or tumor characteristics are associated with a reduced HRQoL. Methods: All patients with a suspected benign/malignant bone tumor (BT), benign soft-tissue tumor (STT), or malignant soft-tissue sarcoma (STS) visiting the Leiden University Medical Center between 2016 and 2020 were invited to complete the Patient-Reported Outcomes Measurement Information System (PROMIS) 29-item profile questionnaire. Mean scores of all included patients and per diagnosis group were compared to mean scores of the general population using one-sample *t*-tests. Results: Overall, patients (*n* = 637) reported statistically significantly worse HRQoL-scores on anxiety (51.3 ± 9.6), pain (55.3 ± 10.1), physical functioning (46.0 ± 9.7), and social functioning (48.1 ± 10.8) with the difference in pain and physical functioning being clinically relevant (based on a 3-point difference on t-metric). HRQoL-scores differed between diagnosis subgroups, i.e., patients with malignant tumors had higher anxiety levels and experienced more pain, where patients with bone tumors had worse physical functioning. Conclusion: The HRQoL of patients with suspected bone and soft-tissue tumors is already affected during the diagnostic process.

## 1. Introduction

Bone and soft-tissue tumors are among the rarest neoplasms in humans, accounting for <1% of all neoplasms arising in the human body [1]. They are a heterogenous group of benign and malignant lesions that may occur at every age at almost any anatomical site, arising from cells of the connective tissue, including muscles, fat, blood vessels, cartilage, and bones. The estimated incidence in Europe per year is around 5/100,000 persons for malignant soft-tissue tumors and 1–2/100,000 persons for malignant bone tumors [2]. In the Netherlands, about 1400 [3] new patients per year are diagnosed with a malignant bone or soft-tissue tumor, with an overall survival of 50–60% at 5 years [2].

With over 100 histological subtypes, and considering the rareness of the disease, diagnosis and treatment of bone and soft-tissue tumors are complex and can be time consuming [4]. Until now, treatment has mainly focused on tumor reduction and improvement in survival. However, in addition to prolonged survival, cancer patients consider the improvement in health-related quality of life (HRQoL) as an important criterion for the treatment of the tumor [5,6,7]. HRQoL is a multidimensional concept that includes the patient’s perspective on the emotional, physical, social, and cognitive domains of well-being The HRQoL of bone and soft-tissue tumor patients has been rarely investigated [8], especially for specific subgroups (benign/malignant) and at different stages of the disease. The few available studies that do investigate HRQoL mostly include a large time period after diagnosis (0–5 years) [9], focus on specific locations [10] or age groups [11], or only include a very small cohort of patients (*n* = 36) [12]. From these studies, we know that bone and soft-tissue tumor patients experience long-lasting physical and psychosocial challenges. However, they do not give insight into the HRQoL challenges patients already experience around the time of diagnosis and if there are differences in HRQoL for specific subgroups of patients. Insight into these challenges would provide further understanding of possibilities for improvement in HRQoL during the diagnostic trajectory, which can improve personalized (after)care tailored to specific bone and soft-tissue tumor subtypes. Moreover, it can optimize shared/informed decision-making. Therefore, the aim of this study was to determine the extent to which patients suspected of having a bone or soft-tissue tumor have reduced HRQoL around the time of diagnosis compared to scores from the general population. In addition, we will assess which patient- and tumor characteristics are associated with clinically relevant reduced HRQoL.

## 2. Methods

### 2.1. Study Design and Participants

This study is an observational cohort study among patients suspected of having a bone or soft-tissue tumor who visited the outpatient oncological orthopedics clinic at the Leiden University Medical Center (LUMC) in the Netherlands between March 2016 and May 2020. The LUMC is one of six sarcoma expertise centers in the Netherlands. All patients (≥16 years) were invited to complete the Patient-Reported Outcomes Measurement Information System (PROMIS) 29-item profile questionnaire [13]. Characteristics for responding patients were retrospectively retrieved from the medical records. As the diagnostic trajectory of bone and soft-tissue tumor patients is complex and studies about this phase are scarce, we allowed a range of 8 weeks around the definitive diagnosis date for patients to be included [14]. Questionnaire data were routinely collected as part of clinical practice and were completed at the outpatient clinic with the help of a student to ensure complete data capture. 

The study protocol was presented to the Medical Ethics Committee of the Leiden University Medical Center, who stated that this study does not fall within the Dutch Medical Research Involving Human Subjects Act (WMO) and therefore waived the need for ethical approval under Dutch law (CME 2022-036).

### 2.2. Health-Related Quality of Life

The PROMIS-29 questionnaire was used to assess patients’ HRQoL status [13]. PROMIS measures are not disease-specific and are therefore universally applicable across various populations [15]. PROMIS measures have also proven to be reliable and valid in a wide range of clinical settings [16,17,18]. PROMIS-29 includes dimensions of mental, physical, and social health across seven domains (anxiety, fatigue, depression, sleep disturbance, pain interference, physical and social functioning). Each domain includes four items on a 5-point scale (e.g., never = 1; always = 5), which means that higher scores indicate more anxiety, fatigue, depression, sleep disturbance, and pain interference (i.e., worse scores), whereas higher scores indicate better physical and social functioning [19]. For each domain, a summary component score can be calculated, standardized to the general US population, with a T-score of 50 as the reference value to indicate the mean of the general population (standard deviation = 10). The Dutch PROMIS-29 manual recommends using these US parameters as no cross-validation study has been carried out in the Netherlands [20]. Based on earlier research among cancer patients, we defined a difference of 3 points on the T-metric as clinically relevant [21]. Therefore, a clinically relevant reduced HRQoL in the domains of anxiety, fatigue, depression, sleep disturbance, and pain interference was defined by a score higher than 53 and a score lower than 47 for the domains of physical and social functioning. 

### 2.3. Variables

To assess which variables may be associated with reduced HRQoL, we retrospectively collected the following data from the medical records: age, sex, tumor location (upper/lower extremity), tumor grade (high-grade), diagnosis group (benign bone tumor (BT), benign soft-tissue tumor (STT), malignant bone tumor (BT), and soft-tissue sarcoma (STS)), as well as whether or not a patient had metastasis at the time of diagnosis. 

### 2.4. Statistical Analyses

First, we wanted to assess whether patients with suspected bone or soft-tissue tumor completing the questionnaires were a selected group of patients by comparing patients’ characteristics (i.e., age, gender) with all other bone or soft-tissue tumor patients visiting the outpatient orthopedic clinic who were not completing the questionnaire. However, we could not specifically select all other patients with bone or soft-tissue tumor based on available data (i.e., both benign and malignant). We only had cancer registry data available for all patients primarily diagnosed with malignant tumors between March 2016 and May 2020 and therefore compared patients with malignant tumors completing and not completing the questionnaires. For patients with malignant bone and soft-tissue tumors, a *t*-test was used to test for differences in age between those completing the questionnaire and those not completing the questionnaire, whereas a chi-square test was used to describe differences in gender. 

Next, we computed the average HRQoL for the specific subdomains among all patients and per diagnosis group (benign bone tumor (BT), malignant BT, benign soft-tissue tumor (STT), and malignant soft-tissue sarcoma (STS)) and compared these with the general population mean score using a one-sample *t*-test. 

Furthermore, we computed the proportion of patients with a clinically relevant reduced HRQoL score and compared patients with and without a reduced HRQoL on patient- and tumor characteristics using a *t*-test (for continuous variables) or chi-square test (for categorical variables). Due to the small number of patients with a metastasis at time of diagnosis (*n* = 10), we could not test for differences between patients with and without a reduced HRQoL in the presence of metastases at the time of diagnosis. All variables that showed a significant univariate relationship (*p* < 0.05) were entered in a multiple logistic regression analysis for each HRQoL domain to examine whether these were associated with reduced HRQoL on that particular domain.

## 3. Results

### 3.1. Patients Characteristics

In total, 637 patients completed the questionnaires and were included. Characteristics of these patients are shown in Table 1. The mean age around the time of diagnosis was 48 years (SD ± 17). Of all patients, 52% were female. Most patients were ultimately diagnosed with a benign bone tumor (62%). The majority of the patients had either a tumor in the lower extremities (61%) or upper extremities (27%). Of the 113 malignant STS and BT patients, 90 patients had a high-grade tumor (80%), 14 patients had a low-grade tumor (12%), and the grade for the remaining 9 patients (8%) could not be classified. Ten patients (1.6%) diagnosed with a malignant tumor had metastasis at time of diagnosis. 

Patients with malignant tumors who completed the questionnaires (*n* = 113) were significantly younger than the total patient population with malignant tumors (*n* = 539) visiting the outpatient clinic of the LUMC (mean ± SD, 46 ± 21 vs. 55 ± 17, *p* < 0.05) but did not differ in gender (%male gender, 48 vs. 50, *p* = 0.057). 

### 3.2. Overall HRQoL and Per Diagnosis Subgroup

Figure 1 shows the PROMIS-29 scores of all patients per HRQoL domain. Compared to the general population mean (T = 50), bone and soft-tissue tumor patients reported on average statistically significantly worse HRQoL scores on anxiety (mean 51.3 ± 9.6), pain interference (55.3 ± 10.1), physical (46.0 ± 9.7), and social functioning (48.1 ± 10.8), and significantly better scores on fatigue (47.1 ± 10.4) and depression (46.4 ± 7.8) (*p* < 0.05). Based on a three-point difference on the T-metric to indicate a clinically relevant reduced HRQoL, this was observed for the domains pain interference and physical functioning.

PROMIS-29 domain scores stratified by patient diagnosis are shown in Figure 2. Patients with a BT, either benign or malignant, reported statistically significant and clinically relevant worse scores on physical functioning (benign 45.8 ± 9.4, malignant 43.4 ± 10) compared with the general population (*p* < 0.05). For soft-tissue tumors, only patients with a malignant tumor reported a clinically relevant worse score on physical functioning (46.9 ± 12). Malignant BT and STS patients reported statistically significant and clinically relevant worse scores on anxiety (BT 55.4 ± 9.7, STS 57.0 ± 8.6) in comparison with the general population (*p* < 0.05), in contrast with benign BT and STT patients, whose anxiety scores were similar to the general population. Pain interference scores are worse in all diagnosis groups, and the difference is clinically relevant, except for benign STT patients. 

### 3.3. Variables Associated with Reduced HRQoL per PROMIS-29 Domain Score

*Anxiety*. Bone and soft-tissue tumor patients with clinically relevant increased anxiety scores were on average older, female, and more often diagnosed with a malignant BT or STS and less often diagnosed with benign BT (Table 2). In multivariate analyses, females had 1.54 odds (95% CI 1.11–2.14) of experiencing clinically relevant increased anxiety (Table 3). Furthermore, being diagnosed with a malignant BT or STS (2.38 (95% CI 1.42–3.99) and 4.76 (95% CI 1.98–11.41), respectively), with benign BT diagnosis as a reference category, increased the odds of experiencing clinically relevant increased anxiety.

*Fatigue.* Patients with clinically relevant increased fatigue were more often female and diagnosed with benign BT, but were less often diagnosed with a benign STT (Table 2). Multivariate analyses showed that female gender was independently associated with increased odds of experiencing clinically relevant increased fatigue (odds ratio 1.57 (95% CI 1.12–2.21)), and only patients diagnosed with a benign STT had lower odds of experiencing clinically relevant increased fatigue (odds ratio 0.62 (95% CI 0.40–0.98)) (Table 4). 

*Depression.* Bone and soft-tissue tumor patients with clinically relevant increased depression scores did not differ in their clinical characteristics except for higher percentages of patients with a malignant BT diagnosis (Table 2). Multivariate regression analysis showed that those BT patients had 1.74 (95% CI 1.04–2.90) higher odds of experiencing increased levels of depression (Table 5). 

*Pain interference*. Bone and soft-tissue tumor patients with clinically relevant increased pain interference scores did not differ in age, gender, tumor location, or grade, except for being diagnosed with a benign STT less often than patients without clinically relevant increased pain interference (Table 2). Multivariate analyses showed that benign STT patients had 0.52 lower odds (95% CI 0.34–0.78) of experiencing clinically relevant increased pain interference (Table 6). 

*Sleep disturbance.* Bone and soft-tissue tumor patients with clinically relevant increased sleep disturbance on average were older, more often female, and more often diagnosed with benign BT and less often diagnosed with a benign STT than patients without clinically relevant increased sleep disturbance. Multivariate analysis showed that age independently increased the odds of experiencing clinically relevant increased sleep disturbance by 1.01 (95% CI 1.00–1.02) per 1 year increase in age (Table 7). Adjusted for the other factors, patients diagnosed with benign STT or malignant BT had lower odds of experiencing clinically relevant increased sleep disturbance (odds ratio 0.55 (95% CI 0.35–0.88) and 0.48 (95% CI 0.27–0.85), respectively). 

*Physical function*. Patients with clinically relevant reduced physical functioning were, on average, older, were more often female, and were diagnosed with a malignant BT more often but less often diagnosed with a benign STT (Table 2). Multivariate analysis showed that the odds of experiencing reduced physical functioning for females were 1.65 higher (95% CI 1.20–2.26) and age increased the odds by 1.01 per year older (95% CI 1.00–1.02), whereas patients diagnosed with an STT had lower odds of experiencing reduced physical functioning by 0.63 (95% CI 0.42–0.96) (Table 8). 

*Social function*. Patients with clinically relevant reduced social functioning were more often diagnosed with a malignant BT and less often with a benign STT compared with patients without clinically relevant reduced social functioning (Table 2). In multivariate analyses with a benign BT diagnosis as reference, a benign STT diagnosis appeared to have a significant association with social functioning, i.e., benign STT patients had 0.55 lower odds (95% CI 0.36–0.85) of experiencing reduced social functioning (Table 9). 

## 4. Discussion

Our results show that bone and soft-tissue tumor patients, on average, had significantly worse HRQoL in the domains of anxiety, pain interference, and physical and social functioning around the time of diagnosis compared with the general population, with the difference in pain interference and physical functioning being clinically relevant. In addition, there were different patterns between diagnosis subgroups, e.g., compared with the general population, we observed clinically relevant higher anxiety scores in patients with malignant tumors and worse physical functioning in patients with bone tumors. Female patients were more likely to experience higher levels of anxiety and fatigue, and limitations in physical functioning. An older age was associated with higher levels of sleep disturbance and limitations in physical functioning. Patients diagnosed with a benign STT were less likely to experience increased levels of fatigue, pain, and sleep disturbance or limitations in physical functioning compared with patients diagnosed with a benign BT. Meanwhile, patients diagnosed with a malignant BT were more likely to experience higher levels of depression, sleep disturbance, and, along with malignant STS patients, higher levels of anxiety. 

### 4.1. Results in Context

Reduced HRQoL scores in bone and soft-tissue tumor patients have been previously reported [9,10,11,12,22], but these studies used different timepoints after treatment, whereas our study results show that the time of diagnosis already has a profound impact on different dimensions of patients’ HRQoL. Reduced physical functioning can be understood from the functional limitations and mobility restrictions resulting from the disease itself, whereas insecurities around a diagnosis with cancer may inevitably lead to persistent emotional and social distress and thereby likely higher anxiety levels. Pain has been shown to be a significant problem in all cancer patients, irrespective of the stage of the disease [23], which highlights the need for adequate pain control right from the moment patients enter the outpatient sarcoma clinic. 

The prospective trial by Parades (2011) is, to our knowledge, the only study that also surveyed HRQoL in bone and soft-tissue tumor patients around the time of diagnosis. However, it had a very small sample size (*n* = 36) and only included patients with a malignant sarcoma. Nevertheless, similar reduced HRQoL scores were found in the domains of physical functioning, social functioning, and pain as in our study. The present study adds to existing literature that at least some bone and soft-tissue tumor patients experience clinically relevant increased anxiety around the time of diagnosis and that female patients and those diagnosed with malignant BT and STS are more likely to experience clinically relevant increased anxiety. Women with cancer having more anxiety symptoms than men is in line with the literature [24]. Interestingly, patients diagnosed with a malignant tumor are clearly more anxious, even during their diagnostic trajectory than patients with a benign tumor, suggesting that these patients may already have been referred to a tertiary referral center in a different way (e.g., as an urgent referral for suspected cancer). Furthermore, while reduced postoperative physical functioning in previous studies was mostly explained to be a consequence of aggressive treatment and their side-effects (e.g., radiotherapy and/or chemotherapy), to which specifically patients with a malignant tumor are exposed [10,12], our study shows that reduced physical functioning already occurs around the time of diagnosis and also in benign tumor patients, so it cannot be entirely explained by treatment or systemic therapies. Prospective longitudinal trials should be conducted to investigate whether HRQoL changes during the course of the disease in different diagnosis subgroups, and if so, in which subdomains.

### 4.2. Strength and Limitations

Strengths of this study include that this is the first study to use PROMIS-29 to assess quality of life among bone and soft-tissue tumor patients. PROMIS-29 measures outcomes that are relevant to all patients, such as pain, anxiety, depression, and physical function, and has the advantage that it can be compared with groups of patients with other diseases or the general population [25]. Although disease-specific measures may provide further insight into patients’ disease-specific symptoms and concerns, given the heterogeneity in histological subtypes, age, and tumor sites and hence the broad treatment landscape, one could question whether all disease-specific items are relevant for every bone or soft-tissue tumor patient [26]. 

Some limitations should be noted. First, this is an observational cohort study of patients suspected to have a sarcoma who are referred to a tertiary sarcoma expertise center. The diagnostic trajectory of these patients is complex due to the many different histological subtypes and their biological behavior which makes the exact moment of diagnosis difficult to determine. Yet, patients referred to a certain expertise center are likely to already be aware they are at risk for a malignant disease, even before the definitive diagnosis, which is why the exact timing of completing the questionnaire is less relevant. Second, our population consists of a large number of bone tumor patients relative to soft-tissue tumor patients due to the study being conducted at the department of oncological orthopedics, while most soft-tissue tumor patients are treated at the department of surgical oncology. The overall results may therefore possibly give a distorted impression in the domain of physical functioning than would be the case if the bone and soft-tissue distribution were more reflective of reality. However, the large number of bone tumor patients in our sample did not have a substantial impact on the other domains, as indicated by the results after stratification. Third, our respondents were significantly younger than the total malignant population, which may indicate an underestimation of increased levels of sleep disturbance and limitations in physical functioning in malignant bone and soft-tissue tumor patients since our results showed that an older age was associated with higher levels of sleep disturbance and limitations in physical functioning. Finally, this study is a single-center study, so the generalization of our results to other settings is uncertain. However, information about HRQoL associated with the uncertainty of the diagnostic trajectory is very scarce and our results provide first important insights into HRQoL scores of patients with bone or soft-tissue tumors. 

### 4.3. Implications

Pre-treatment information about HRQoL is rare in cancer patients; our results provide first insights into baseline scores of bone and soft-tissue tumor patients. Insight gained from these scores can be used to develop interventions aimed at improving patients’ HRQoL and should ideally be personalized and of a multidisciplinary nature to cover all the different HRQoL domains. For example, given the results of our study, female patients are at risk of experiencing higher levels of anxiety. Thus, patients’ anxiety levels can become a topic to discuss during the clinical encounter, and if limitations are observed, paramedical support might be recommended. Interventions as such are particularly important in the initial phase of the disease, as patients with an initially worse HRQoL are at higher risk for a poorer HRQoL later on [12]. To monitor which patients experience limitations in HRQoL and might need additional care (i.e., pain medication, paramedical support), an HRQoL questionnaire must become part of standard sarcoma care. 

## 5. Conclusions

This study shows that patients visiting the outpatient clinic with a bone or soft-tissue tumor report worse HRQoL compared with the general population with regard to pain and physical functioning already around the time of diagnosis. This information, together with the identification of specific patient or tumor characteristics associated with reduced HRQoL during the diagnostic trajectory, is important as it allows clinicians and nurses to adjust the care given to individual cases. 

## Figures and Tables

**Figure 1 cancers-15-02804-f001:**
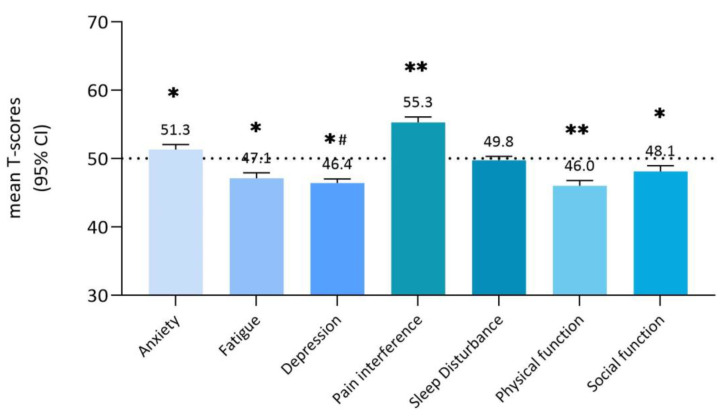
PROMIS-29 domain scores of bone and soft-tissue tumor patients (*n* = 637) in comparison with the general population mean (T = 50). The seven PROMIS domains are represented on the *x*-axis. Mean PROMIS T-scores are represented on the *y*-axis. A clinically important difference is approximately ± 3 point difference on T-metric. * *p* < 0.05, ** clinically and significantly (*p* < 0.05) worse than reference, *# clinically and significantly (*p* < 0.05) better then reference.

**Figure 2 cancers-15-02804-f002:**
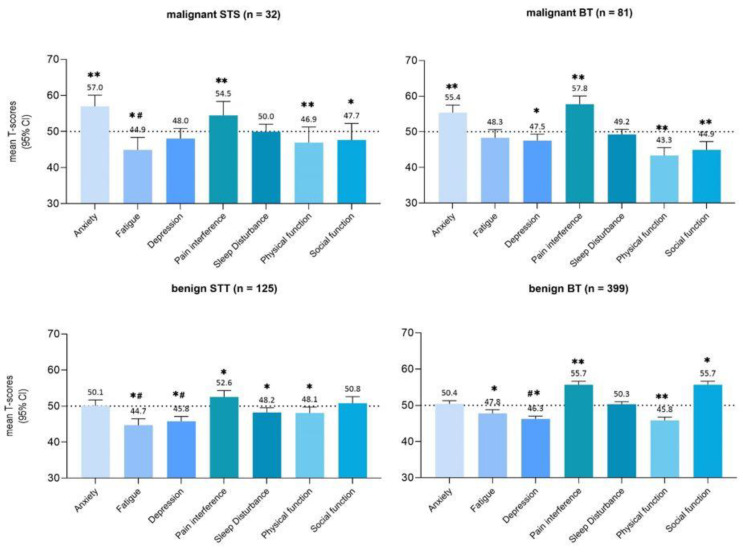
PROMIS-29 domain scores stratified by patients’ diagnosis compared to general population mean (T = 50). The seven PROMIS domains are represented on the x-axis. Mean PROMIS T-scores are represented on the y-axis. A clinically important difference is approximately ±3 point difference on T-metric. * *p* < 0.05, ** clinically and significantly (*p* < 0.05) worse than reference, *# clinically and significantly (*p* < 0.05) better then reference.

**Table 1 cancers-15-02804-t001:** Characteristic of patients diagnosed with a bone or soft-tissue tumor around the time of diagnosis.

Characteristics	Bone and Soft-Tissue Tumor Patients (*n* = 637)
Mean age y (±SD)	48 ± 17
Sex Female	335 (52%)
Diagnosis Malignant STS (Myxo)fibrosarcoma Synovial sarcoma (Myoid)liposarcoma UPS/NOS Angiosarcoma Others Malignant BT Chondrosarcoma Osteosarcoma Multiple myeloma Chordoma Ewing Sarcoma Others Benign STT TGCT ALT/Lipoma Fibroma Myxoma Schwannoma Others Benign BT ACT/CS 1, Enchondroma Fibrous dysplasia Osteoid osteoma Osteochondroma Giant cell tumor Others	32 (5%) * 10 (31.3%) ** 3 (9.4%) 3 (9.4%) 3 (9.4%) 2 (6.2%) 11 (34.3%) 81 (13%) 41 (50.6%) 13 (16%) 6 (7.4%) 5 (6.2%) 4 (4.9%) 12 (14.8%) 125 (19%) 34 (27.2%) 29 (23.2%) 10 (8%) 9 (7.2%) 8 (6.4%) 35 (28%) 399 (62%) 182 (45.6%) 49 (12.3%) 43 10.8%) 34 (8.5%) 18 (4.5%) 73 (18.3%)
Tumor site Head and neck Thorax/spine Upper extremities Lower extremities	8 (1.2%) 70 (11%) 171 (27%) 394 (61%)
Grade Low grade (I) High grade (II-III) Unable to classify	*n* = 113 ^ 14 (12%) 90 (80%) 9 (8.0%)
Metastasis at diagnosis Yes	10 (1.6%)

BT, bone tumor. SD, standard deviation. STS, soft tissue sarcoma. STT, soft tissue tumor. UPS/NOS, undifferentiated pleomorphic sarcoma/sarcoma not otherwise specified. TGCT, tenosynovial giant cell tumor, ALT, atypical lipomatous tumor. ACT, atypical cartilaginous tumor. CS, chondrosarcoma. * percentage of total population, ** percentage of subgroup, ^ only to be determined in malignant tumors.

**Table 2 cancers-15-02804-t002:** Comparison of patients with and without reduced HRQoL per domain on patient and tumor characteristics.

	Anxiety			Fatigue			Depression			Pain Interference			Sleep Disturbance			Physical Function			Social Function		
Clinically relevant reduced HRQoL *	Yes N = 304	No N = 333	*p*	Yes N = 211	No N = 426	*p*	Yes N =153	No N = 484	*p*	Yes N = 412	No N = 225	*p*	Yes N = 201	No N = 436	*p*	Yes N = 337	No N = 300	*p*	Yes N = 271	No N = 366	*p*
Age mean + SD	49.9 ± 15.8	46.3 ± 17.1	**0.01**	48.0 ± 17.0	48.1 ± 15.8	0.93	49.6 ± 15.7	47.5 ± 16.8	0.19	48.2 ± 16.5	47.7 ± 16.7	0.71	49.9 ± 16.5	47.1 ± 16.6	**<0.05**	49.6 ± 16.4	46.2 ± 16.7	**0.01**	48.4 ± 16.1	47.7 ± 17.0	0.58
Sex % Female	56.6	48.0	**0.03**	60.2	48.1	**0.00**	56.9	50.6	0.18	54.9	47.1	0.06	58.2	49.3	**0.04**	58.2	45.3	**0.00**	54.6	50.3	0.28
Diagnosis % Malignant STS % Malignant BT % Benign STT % Benign BT	8.2 17.4 16.8 57.6	2.1 8.4 22.2 67.3	**0.00** **0.00** 0.08 **0.01**	2.8 13.3 15.2 68.7	6.1 12.4 21.8 59.6	0.08 0.77 **<0.05** **0.03**	6.5 18.3 14.4 63.2	4.5 11.0 21.3 60.8	0.33 **0.02** 0.06 0.59	4.6 14.1 15.8 65.5	5.8 10.2 26.7 57.3	0.52 0.16 **0.00** **0.04**	6.5 9.0 14.9 69.7	4.4 14.4 21.8 59.4	0.26 0.05 **0.04** **0.01**	3.9 15.4 16.3 64.4	6.3 9.7 23.3 60.7	0.15 **0.03** **0.03** 0.33	5.2 15.9 14.0 64.9	4.9 10.4 23.8 60.9	0.89 **0.04** **0.00** 0.30
Location % Upper Ex % Lower Ex	23.7 62.2	29.1 60.4	0.12 0.64	25.1 61.6	27.2 61.0	0.57 0.89	23.5 64.7	27.5 60.1	0.34 0.31	26.9 60.2	25.8 63.1	0.75 0.47	24.4 64.2	27.5 59.9	0.40 0.30	24.9 62.9	28.3 59.3	0.33 0.36	25.8 59.0	27.0 62.8	0.73 0.33
Grade % High grade	83.3	71.4	0.15	73.5	82.3	0.29	86.8	76.0	0.18	83.1	72.2	0.18	83.9	78.0	0.49	83.1	75.0	0.29	75.4	83.9	0.26

* Scores ≥ 53 in the domains of anxiety, fatigue, depression, pain interference, and sleep disturbance are defined as clinically relevant reduced HRQoL, and scores ≤47 on physical and social functioning. Significant differences (*p* < 0.05) are in bold.

**Table 3 cancers-15-02804-t003:** Logistic regression analysis showing the independent associations of sex, age, and diagnosis with clinically relevant increased anxiety and sex, age, and diagnosis.

	*p*	Odds Ratio	95% CI for Odds Ratio	
			Lower	Upper
Sex ^	**0.01**	1.54	1.11	2.14
Age at diagnosis	0.08	1.01	1.00	1.02
Benign BT		1		
Benign STT	0.50	0.87	0.57	1.32
Malignant BT	**<0.01**	2.38	1.42	3.99
Malignant STS	**<0.01**	4.76	1.98	11.41

^ Sex is for females compared to males; Significant differences are in bold.

**Table 4 cancers-15-02804-t004:** Logistic regression analysis showing the independent associations of sex and diagnosis with clinically relevant increased fatigue.

	*p*	Odds Ratio	95% CI for Odds Ratio	
			Lower	Upper
Sex ^	**0.01**	1.57	1.12	2.21
Benign BT		1		
Benign STT	**0.04**	0.62	0.40	0.98
Malignant BT	0.95	0.98	0.59	1.63
Malignant STS	0.08	0.44	0.18	1.11

^ Sex is for females compared to males; Significant differences are in bold.

**Table 5 cancers-15-02804-t005:** Logistic regression analysis showing the independent associations of diagnosis with clinically relevant increased depression.

	*p*	Odds Ratio	95% CI for Odds Ratio	
			Lower	Upper
Benign BT		1		
Benign STT	0.18	0.70	0.42	1.18
Malignant BT	**0.04**	1.74	1.04	2.90
Malignant STS	0.31	1.50	0.68	3.27

Significant differences are in bold.

**Table 6 cancers-15-02804-t006:** Logistic regression analysis showing the independent associations of diagnosis with clinically relevant increased pain interference.

	*p*	Odds Ratio	95% CI for Odds Ratio	
			Lower	Upper
Benign BT		1		
Benign STT	**<0.01**	0.52	0.34	0.78
Malignant BT	0.49	1.21	0.71	2.04
Malignant STS	0.34	0.70	0.34	1.46

Significant differences are in bold.

**Table 7 cancers-15-02804-t007:** Logistic regression analysis showing the independent associations of sex, age, and diagnosis with clinically relevant increased sleep disturbance.

	*p*	Odds Ratio	95% CI for Odds Ratio	
			Lower	Upper
Sex ^	0.09	1.35	0.95	1.90
Age at diagnosis	**0.01**	1.01	1.00	1.02
Benign BT		1		
Benign STT	**0.01**	0.55	0.35	0.88
Malignant BT	**0.01**	0.48	0.27	0.85
Malignant STS	0.63	1.20	0.57	2.56

^ Sex is for females compared to males; Significant differences are in bold.

**Table 8 cancers-15-02804-t008:** Logistic regression analysis showing the independent associations of sex, age, and diagnosis with clinically relevant reduced physical functioning.

	*p*	Odds Ratio	95% CI for Odds Ratio	
			Lower	Upper
Sex ^	**<0.01**	1.65	1.20	2.26
Age at diagnosis	**0.02**	1.01	1.00	1.02
Benign BT		1		
Benign STT	**0.03**	0.63	0.42	0.96
Malignant BT	0.17	1.43	0.86	2.39
Malignant STS	0.14	0.57	0.27	1.20

^ Sex is for females compared to males; Significant differences are in bold.

**Table 9 cancers-15-02804-t009:** Logistic regression analysis showing the independent associations of diagnosis with clinically relevant reduced social functioning.

	*p*	Odds Ratio	95% CI for Odds Ratio	
			Lower	Upper
Benign BT		1		
Benign STT	**0.01**	0.55	0.36	0.85
Malignant BT	0.14	1.43	0.89	2.32
Malignant STS	0.97	0.99	0.48	2.04

Significant differences are in bold.

## Data Availability

The data that support the findings of this study are available on request from the corresponding author, upon reasonable request.

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
