# Peer review of "Health-Related Quality of Life of Bone and Soft-Tissue Tumor Patients around the Time of Diagnosis"

_cancers, 2023, doi:10.3390/cancers15102804_

Round 1

Reviewer 1 Report

The paper regarding “Health-related quality of life of bone and soft-tissue tumor patients around the time of diagnosis.” is presented. In this study, the authors used patients with a suspected benign/malignant bone tumor (BT), benign soft-tissue tumor (STT) or malignant soft-tissue sarcoma (STS) and assessed health-related quality of life (HRQoL) of bone and soft-tissue tumor patients around the time of diagnosis and explores which patient- or tumor characteristics are associated with reduced HRQoL.  They found taht HRQoL-scores differed between diagnosis subgroups, i.e. patients with malignant tumors had higher anxiety levels and experienced more pain, whereas patients with bone tumors had worse physical functioning. The study is well-presented and potentially interesting.

There are some possible issues

In table 1, it would be informative to subclassify bone tumours into which types of bone tumors such as osteoclastoma (giant cell tumour of bone), osteosarcoma, or tumour metastasis to bone ?  It might include an additional table for this?

In terms of diagnosis, more recent work has used RNAseq for further precision diagnosis (for example, PMID: 34367994; PMID: 34504794).  It would be relevant to discuss this aspect for a clear diagnosis rather than a suspected diagnosis. 

Can you include diagnosis at the histology level?

There are some typos, for example:

diagnosis and treatment of bone and soft-tissue tumors is ?? are ?

Insight in these challenges?? Insight into these challenges?

Author Response

Dear reviewer,

Thank you for your careful reading of our manuscript and constructive comments. We think these have considerably improved the manuscript. Please find attached our response. The line numbers refer to the revised document with track changes.

Yours sincerely,

Anouk Kruiswijk

Reviewer 2 Report

Manuscript ID: cancers-2258276

Title: Health-related quality of life of bone and soft-tissue tumor patients around the time of diagnosis

This study presents unique data on a rare population of patients and has a lot to add to the literature. However, the way the data are evaluated and displayed make it challenging to interpret. I would recommend working with a statistician to better display the data for ease of interpretation of the reader.  

Major Points:

1.     Page 3, line 78. The sentence discussing the routinely collected data from the outpatient orthopedic clinic is confusing. Is this the general population referred to in section 3.2 if results? It is not clear what the “general population” cohort was obtained from.

2.     Table 2is confusing and needs some clarification. How was the cutoff of 53 determined to be the cutoff used in the tables? Is this table comparing characterisitcs of the PROMIS elements above and below the mean? The way these data are displayed are very confusing to interpret. 

Minor Points:

1.     Table 2 is out of order in the manuscript.

Top of page 6. There is a random sentence that starts with “Scheme 0” Is that associated with the table or the results?

Author Response

Dear reviewer,

Thank you for your careful reading of our manuscript and constructive comments. We think these have considerably improved the manuscript. Please find attached our detailed response. The line numbers refer to the revised document with track changes. 

Yours sincerely,

Anouk Kruiswijk

Reviewer 3 Report

The reason for performing the study is not clearly stated. One can reasonably assume that diagnosis of any major disorder results in anxiety and alters physical and social functioning. Therefore, the value of the findings is not clear.

Other points to be clarified includes the statement that “Medical Ethical Committee had waived for ethical approval”. This is highly unusual, and approval must have been required. The method for selection of patients and controls is also nonconventional and is not clear if this has had any effect on the results. Predominance of younger patients noted in the study likely has potential to alter the findings. The ratio of patients with bone tumors, as opposed to those with soft tissue is high, distorting the data and making interpretation of the results difficult. The range of time to completion of the questionnaire appears to have varied among patients. The degree of the patient’s knowledge of their grade of their disease presumably was not evaluated. The authors state that “For each domain a summary component score can be calculated, standardized to the general US population”. This action was taken since, according to the authors ”no cross-validation study has been carried out in the Netherlands”. Comparison of two different and distinct population may not be justified. It is of note that these matters were not significantly discussed in the manuscript. Some references do not correspond to the subject discussed.

There are several typographical mistakes in the manuscript and the English language writing can be significantly improved.

Author Response

Dear reviewer,

Thank you for your careful reading of our manuscript and constructive comments. We think these have considerably improved the manuscript. Please find attached our detailed response. The line numbers refer to the revised document with trackchanges.

Your sincerely,

Anouk Kruiswijk

Round 2

Reviewer 2 Report

addressed for my previous comments.

Reviewer 3 Report

The response of the authors is appreciated.